# Microstructured Polymeric Fabrics Modulating the Paracrine Activity of Adipose-Derived Stem Cells

**DOI:** 10.3390/ijms241210123

**Published:** 2023-06-14

**Authors:** Federica Grilli, Ennio Albanesi, Beatriz Pelacho, Felipe Prosper, Paolo Decuzzi, Daniele Di Mascolo

**Affiliations:** 1Laboratory of Nanotechnology for Precision Medicine, Fondazione Istituto Italiano di Tecnologia, 16163 Genoa, Italy; federicagcor.92@postecert.it (F.G.); paolo.decuzzi@iit.it (P.D.); 2Department of Informatics, Bioengineering, Robotics, and Systems Engineering (DIBRIS), University of Genoa, 16145 Genoa, Italy; 3Department of Neuroscience and Brain Technologies, Fondazione Istituto Italiano di Tecnologia, 16163 Genoa, Italy; ennio.albanesi@iit.it; 4Laboratory of Regenerative Medicine, Center for Applied Medical Research, University of Navarra, 31008 Pamplona, Spain; bpelacho@unav.es (B.P.); fprosper@unav.es (F.P.); 5Instituto de Investigación Sanitaria de Navarra (IdiSNA), 31008 Pamplona, Spain; 6Department of Electrical and Information Engineering, Politecnico di Bari, 70126 Bari, Italy

**Keywords:** tissue engineering, scaffolds, adipose-derived stem cells, spheroids, paracrine activity

## Abstract

The deposition of stem cells at sites of injury is a clinically relevant approach to facilitate tissue repair and angiogenesis. However, insufficient cell engraftment and survival require the engineering of novel scaffolds. Here, a regular network of microscopic poly(lactic-co-glycolic acid) (PLGA) filaments was investigated as a promising biodegradable scaffold for human Adipose-Derived Stem Cell (hADSC) tissue integration. Via soft lithography, three different microstructured fabrics were realized where 5 × 5 and 5 × 3 μm PLGA ‘warp’ and ‘weft’ filaments crossed perpendicularly with pitch distances of 5, 10 and 20 μm. After hADSC seeding, cell viability, actin cytoskeleton, spatial organization and the secretome were characterized and compared to conventional substrates, including collagen layers. On the PLGA fabric, hADSC re-assembled to form spheroidal-like structures, preserving cell viability and favoring a nonlinear actin organization. Moreover, the secretion of specific factors involved in angiogenesis, the remodeling of the extracellular matrix and stem cell homing was favored on the PLGA fabric as compared to that which occurred on conventional substrates. The paracrine activity of hADSC was microstructure-dependent, with 5 μm PLGA fabric enhancing the expression of factors involved in all three processes. Although more studies are needed, the proposed PLGA fabric would represent a promising alternative to conventional collagen substrates for stem cell implantation and angiogenesis induction.

## 1. Introduction

After an injury, many biological pathways become activated and respond to the harmful stimulus. Stem-cell-based therapy is an effective approach to facilitate wound healing and tissue regeneration, primarily via capitalizing on the paracrine activity of stem cells, such as the release of a variety of cytokines, chemokines, and growth factors that activate host cells, promote the homing of other cells to the site of injury and, thus, enhance the host’s intrinsic repair mechanisms [1,2,3]. The beneficial effect of this cell therapy ultimately depends on the number of implanted cells at the target site, their viability, their engraftment into damaged tissue and their ability to promote tissue regeneration. Human Adipose-Derived Stem Cells (hADSC) represent an interesting stem cell candidate for tissue regeneration as they secrete bioactive molecules [4], regulate the local immune response, and compared with stem cells harvested via bone marrow aspiration, hADSC from fat tissue are easily and abundantly available and carry a relatively lower donor site morbidity since they can be harvested using minimally invasive techniques (i.e., liposuction) [5,6]. The trophic and immunomodulatory activities exerted by hADSC would suggest that hADSC may serve as site-regulated ‘drugstores’ upon implantation [7]. Nevertheless, the challenges associated with preserving stem cells in a culture and facilitating their engraftment in the damaged tissue remain [8,9]. Additionally, another challenge in clinical practice concerns the poor long-term maintenance of stem cell functions [10,11].

Recently, biomaterials have been proposed and developed to overcome these bottlenecks and support cell attachment and retention at the desired site [3,9]. Specifically, advances in biomaterial science combined with a more detailed knowledge about extracellular matrix (ECM) biology and the role of environmental factors in tissue formation and shaping have led to the development of biocompatible scaffolds tailored to provide appropriate structural support and, in some cases, biological and mechanical stimuli to promote tissue regeneration [12,13,14]. Naturally derived polymeric materials, including polypeptides (e.g., collagen) and polysaccharides (e.g., hyaluronic acid), have been extensively explored as scaffolding materials [15,16]. Native ECM can also serve this function after decellularization [8].

In addition to their modest mechanical strength, naturally derived polymers are typically digested over time by enzymes, following a complex and often not controllable degradation kinetics process [8]. Differently, synthetic polymers can be manufactured with higher reproducibility, a wider range of mechanical properties and degradation kinetics [17], and a surface amenable to chemical modification [18,19,20]. Further, synthetic polymers are cheaper than natural polymers are and have longer storage time [18]. The most widely used and clinically approved synthetic polymers are poly (ε-caprolactone) (PCL), poly (lactic acid) (PLA), poly (glycolic acid) (PGA), and its copolymers poly (lactic-co-glycolic acid) (PLGA) [18]. Polyethylene Terephthalate (PET) is also a promising scaffolding biomaterial, which has been used, for example, to realize vascular grafts [21,22]. However, PET is a non-biodegradable linear polyester with notable mechanical properties and biological features, including biocompatibility, biostability, and the promotion of tissue growth [22,23]. Despite that, the high hydrophobicity of PET restricts its application in medicine [24]. On the downside, synthetic polymers do not support cell attachment as natural polymers and may trigger a local immune response and toxicity [25]. To overcome these limitations, hybrid scaffolds combining natural polymers (i.e., collagen, gelatin, or calcium phosphate) and synthetic polymers have been proposed to improve hydrophilicity, cell attachment, and biodegradability [26]. The success of a scaffold largely depends on its proper integration at the implantation site and the realization of conditions supporting the implanted cells, while favoring the intimate interaction with resident and infiltrating cells [8].

Moreover, based on the importance of stem cell transplantation in clinical practice, multiple laboratories have extensively tested different culturing conditions and microenvironments to favor cell nesting and replication, as well as guide cell differentiation. A spheroid-based culture system helps establish a physico-chemical microenvironment such as that in vivo by facilitating cell–cell and cell–matrix interactions, thus overcoming the limitations of conventional monolayer cell cultures [27]. In addition, spheroids of mesenchymal stem cells (MSCs) tend to preserve their intrinsic phenotypic properties via cell–extracellular matrix interactions [27]. However, spheroid cell culture systems are also characterized by having hypoxic and necrotic cores [27]. Several strategies are available to support the arrangement of cells into spheroids, including the use of biocompatible hydrogel scaffolds made of alginate, fibrin, collagen, or hyaluronic acid [28,29,30]; thin biocompatible films made of chitosan or graphene [27,31,32]; micrometric particles whose incorporation into spheroids controls the cell culture condition [33,34,35]. In particular, hydrogels have been extensively used to replicate the typical in vivo environment to improve cell viability, preserve stemness, and favor angiogenesis. On the other hand, biofilms have been shown to increase adhesion and proliferation, while preserving the differentiation potential. Finally, microparticles have been documented to control mechano-transduction mechanisms within the spheroid to eventually improve viability and proliferation [27].

Here, a microfabricated network of poly(lactic-co-glycolic acid) filaments, organized as orthogonal warps and wefts in a polymeric fabric, is demonstrated and tested for its biocompatibility with hADSC. Taking advantage of the accurate control in size, shape, and surface proprieties offered by top-down fabrication approaches such as soft lithography [36], three different configurations of PLGA fabrics were realized, with warp (or weft) pitch distances of 5, 10, and 20 μm, respectively. The resulting PLGA microstructured fabrics are biodegradable and mechanically flexible to favor cell adhesion and tissue integration [36,37]. A critical comparison with conventional substrates for in vitro cell culturing (2D Petri dish) and in vivo cell transplantation (2D collagen Type I layer) is also presented.

## 2. Results

### 2.1. Fabrication and Characterization of the Microfilament Networks

A top-down fabrication strategy (Figure 1) based on soft lithographic techniques was employed to realize three different microfilament network configurations, following protocols described by the authors^1,2,6^. The process started with the realization of a silicon master template via direct laser writing (DLW) carrying a regular matrix of square pillars with a height of 5 μm, an edge length of 5, 10, or 20 μm, and a separation distance of 3 or 5 μm depending on the microfilament network configuration (Figure 1A). This master template was replicated in an intermediate PDMS template (Figure 1B) and, eventually, into a sacrificial PVA template (Figure 1C), which accurately reproduces the regular matrix of square pillars such as those in the original silicon template. To generate the microfilament network, the empty space between adjacent pillars in the sacrificial PVA template is carefully filled with a solution of PLGA, which was dissolved in rapidly evaporating organic solvent acetonitrile (Figure 1C). Finally, the PVA template was removed upon exposure to water releasing a network of PLGA microscopic filaments whose nodes, edges, and openings correspond to the original ridges within the matrix of pillars (Figure 1D). In the current work, three network configurations were realized presenting different separation distances (pitch) of 5, 10, or 20 μm between the adjacent PLGA microscopic filaments. Additionally, the microscopic filaments cross section was 5 × 5 μm for the first two configurations and 3 × 5 μm for the 20 μm network.

In Figure 2, scanning electron microscopy (SEM) images show the PVA sacrificial templates, highlighting the regular square pillar matrix (left column) and the PLGA microfilament networks resulting from PVA dissolution in water (central column). Fluorescent microscopy images document the uniform distribution of the natural green-fluorescent molecule curcumin, which is dispersed within the PLGA microfilament network during fabrication (right column). These SEM and fluorescent microscopy images document the regular structure of the PLGA microfilament network and the accurate control on the network geometry via the proposed top-down fabrication strategy.

### 2.2. Stem Cells Clustering on PLGA Microfilament Networks

Human Adipose-Derived Stem Cells (hADSC) were used to perform all the in vitro cell experiments. At predetermined time points, namely 24, 48, and 72 h, the interaction between hADSC and the microfilament networks was assessed in terms of the cells’ propensity to organize in large clusters, spheroids, and compared to regular cell culture plastic dishes (2D Petri dish) and planar collagen type I scaffolds (2D collagen layer). On all the microfilament networks, hADSC formed spheroidal aggregates, as reflected by the bluish (fluorescent) spots shown in Figure 3A(iii,iv). Differently, the same cells were observed to adhere and spread on Petri dishes and collagen scaffolds, forming quasi-continuous monolayers (Figure 3A(i,ii)). Interestingly, as shown in Figure 3B, the microfilament network tended to wrap around the cellular aggregates, undergoing stretching and distortions, favoring a more intimate interaction with the adipose stem cells. The number, size, and shape of hADSC spheroids was characterized for all the three different microfilament network configurations at three predetermined time points, as detailed in Figure 3C–E. By analyzing multiple fluorescent microscopy images, the number of spheroids counted on top of the 5 μm microfilament network appeared to be higher than those for the other configurations at 24 h (Figure 3C). Namely, at this time point, 547 ± 307 spheroids were formed on the 5 μm network, as opposed to 173 ± 82 and 247 ± 176 for the 10 μm and 20 μm configurations, respectively. However, the difference is not statistically significant. At 72 h, 133 ± 45, 108 ± 60, and 164 ± 132 spheroids were counted on the 5 μm, 10 μm, and 20 μm configurations, respectively. In general, the number of spheroids reduced with time for all the microfilament networks, suggesting either a reassembly of the original aggregates or their detachment. Note that only firmly adhering spheroids are accounted for in Figure 3C–E. The size of the spheroids was assessed via measuring their perimeters computed as the length of the contour of each spheroid. Only modest changes in the size of the spheroids were documented over time and among the three different microfilament network configurations. The hADSC spheroids on the 20 μm microfilament network appeared to be slightly smaller than the others did (Figure 3D). At 24 h, spheroids on the 5 μm network returned an average size of 473.1 ± 253.3 μm, which remained almost constant over time (i.e., 476.0 ± 250.6 μm at 72 h). On the other hand, the spheroids had average sizes of 499.9 ± 199.3 μm and 423.7 ± 186.7 μm for the 10 and 20 μm configurations at 24 h, which became 441.3 ± 185.1 μm, and 353.4 ± 144.3 μm at 72 h, respectively. Interestingly, over time, hADSC aggregates appeared to rearrange, reducing their aspect ratio and adopting a more spheroidal shape. For all the microfilament networks, the aspect ratio, which is defined as the ratio between the major axis and the minor axis, tended to 1, and statistically significant differences were documented over time (Figure 3E and Appendix A Appendix A).

### 2.3. Stem Cell Viability on PLGA Microfilament Networks

Flow cytometry analyses were conducted to assess the percentage of live, apoptotic, and dead cells in the aggregates. Annexin V-FITC was used to identify apoptotic cells, Propidium Iodide was used to recognize dead cells, while the live cells had no fluorescent signal. hADSC viability was compared to that of the same cells plated on plastic Petri dishes and on 2D collagen type I scaffolds (control experiments). The hADSC cultured on the microfilament network presented with a percentage of live cells ranging between 70 and 80%, about 20% apoptotic cells, and 5% dead cells for all the microfilament networks, regardless of the predetermined time point (Figure 4 and Appendix A Appendix A). These values were comparable to those observed for the 2D collagen type I scaffolds. On the other hand, more viability was documented on 2D Petri dish with 90% live cells for all three time points. Indeed, the major difference between the microfilament networks and the 2D substrates is that on the former ones, hADSC tend to form spheroids rather than growing in monolayers. The different spatial cell organization is expected to affect the distribution of nutrients to the core of the hADSC aggregates, possibly explaining the lower viability as compared to that of the Petri dish, although it is similar to other scaffolds (i.e., collagen) in which this 3D architecture was not present.

### 2.4. Actin Reorganization in Stem Cells on PLGA Microfilament Networks

Given the different spatial organization of cells seeded on 2D substrates (Petri dishes and collagen scaffolds) as opposed to those on the microfilament networks, the differences in the arrangement of actin filaments within the cytoskeleton of adipose stem cells was also investigated. Cells were stained with Phalloidin-488 and observed under a confocal fluorescent microscope. Significant differences were detected in the actin organization, depending on the type of culturing substrate. First, actin filaments would preferentially assemble following a nonlinear structure for all three microfilament network configurations, which is in striking contrast with the 2D substrates that favor the linear organization of actin. This is documented in the microscopy images in Figure 5A. On the 2D substrates, cells appeared to be elongated and well spread, with actin filaments running along the major axis of the cell body (Figure 5A(i,ii)). The actin filament organization is more complex in the hADSC spheroids formed on the microfilament networks, whose strands carry molecules of red fluorescent dye Rhodamine B to enhance the contrast with green Phalloidin-488 used to stain cells (Figure 5A(iii,iv)). More specifically, on the microfilament networks, actin formed ring-like structures at the points of anchorage with the PLGA microscopic filaments. Following the thorough characterization of fluorescent images, over 60% of the actin filaments of the cells cultured on 2D substrates were linearly organized, as opposed to less than 10% for the cells deposited over the PLGA microscopic filament networks (Figure 5B). The 10 μm configuration appeared to have a slightly higher propensity for linearly organized actin filaments (>5%), which is in contrast with the other two configurations, where less than 2% of the actin fibers were linearly organized. However, no statistically significant difference was detected among the three microfilament network configurations.

### 2.5. Secretome Analysis of Inflamed Stem Cells on Microfilament Networks

To determine the influence of the substrate on the paracrine activity of hADSC, supernatants were collected and analyzed using a cytokine array at 48 h post-cell seeding. The membrane-based cytokine array presents antibodies printed on a membrane that are capable of recognizing 80 different analytes (cytokines, chemokines, or growth factors) from the supernatant. Following biotin and streptavidin binding, the membranes were analyzed via chemiluminescence. ImageJ was used to quantify the size of the dark dots. The baseline, derived as the average value of the negative control dots, was subtracted from all the measured quantities. Finally, all the obtained values were normalized according to the average value of the positive control dots. Bigger and darker dots indicate that there are more proteins in the sample. Among the 80 targets detected with the kit, particular attention was given to those involved in angiogenesis (vascular endothelial growth factor—VEGF; platelet-derived growth factor-BB—PDGF-BB), stem cell proliferation (fibroblast growth factor family—FGF), cell survival (endothelial growth factor—EGF), inflammation modulation (interleukin-10—IL-10; transforming growth factor β1—TGF-β1), ECM remodeling (tissue inhibitor metalloproteinase 2—TIMP-2), stem cell homing (granulocyte colony stimulating factor—GCSF) and promoting chemotaxis (regulated upon activation, normal T cell expressed and secreted—RANTES), with the idea that the paracrine effect of hADSC could be modulated using the specific substrate. Figure 6 shows the levels of expression of all these relevant chemokines, cytokines, and growth factors. Interestingly, the 5 μm microfilament network configuration had a significantly higher VEGF secretion level and a trend of increasing PDGF-BB release, thus potentially favoring an angiogenic response as compared to the responses of the other configurations and the 2D substrates, which were used as controls. Specifically, a statistically significant difference (*p* < 0.01) was detected when the 5 μm configuration was compared with the 2D Petri dish and the collagen type I scaffold. Additionally, the 5 μm microfilament network improved the secretion of TIMP-2 by 60% (*p* < 0.01), which is involved in ECM remodeling and wound healing, as compared to that of the 2D collagen scaffolds. Additionally, this configuration raised the secretion of GCSF, which is involved in stimulating the bone marrow to deliver stem cells into the bloodstream, by 20-fold (*p* < 0.05) as compared to the 2D Petri dish. Notably, the 5 μm configuration also elicited the production of TGF-β1, IL-10, and RANTES relative to that of the 2D substrates, demonstrating its possible contribution to immuno-modulation and chemotaxis; although, no statistically significant differences were found among the groups. Regarding the 10 μm microfilament network, a statistically significant difference was found for VEGF secretion relative to that of the 2D Petri dish group (*p* < 0.05), demonstrating its involvement in angiogenesis. This configuration also enhanced the release of FGF family and EGF, which are implicated in stem cell proliferation and cell survival, respectively, as compared to that in the 2D collagen condition; although, these results are without statistically significant differences. Moreover, a statistical difference was found between the 10 μm microfilament network and the 2D collagen scaffolds (*p* < 0.05) for the secretion of the TIMP-2 factor. Finally, the 20 μm microfilament network was mainly implicated in ECM remodeling. Indeed, it enhanced TIMP-2 production by 40% (*p* < 0.05) as compared to that of the 2D collagen scaffold. Moreover, it slightly increased the secretion of IL-10 and RANTES as compared to that of the 2D collagen scaffold, demonstrating its contribution to the immune system regulation and chemotaxis; although these results are without statistical differences.

## 3. Discussion

The flexible microfilament network was designed to realize a physical support for stem cell transplantation and stimulate tissue regeneration post-injury. The presented top-down fabrication strategy can precisely tailor the geometrical and mechanical features of microfilament networks, where the width and thickness of the PLGA filaments, as well as their separation distance and arrangement, can be independently tuned during the fabrication process by simply using a different silicon master template.

The micrometric PLGA fibers are very thin and flexible and are arranged to form wide regular openings of 5 × 5, 10 × 10, and 20 × 20 µm. This unique architecture allows the fabrics to be extremely flexible and establish a close interaction with the stem cells, such as, for instance, those documented in a previous manuscript by the authors [36]. The flexibility per se without the openings (i.e., geometry) would not be enough to establish such an intimate interaction with the cultured cells.

Interestingly, the microfilament networks allow hADSC to spontaneously form 3D spheroidal-like structures.

The relationship between the number of seeded cells and the number of spheroids formed over time on top of the three microfilament networks depends on different factors. The number of spheroids detected at 24 h was higher in the 5 μm PLGA fabric relative to those in the other configurations, probably because the smaller openings could more efficiently retain the stem cells and prevent them from dropping on the bottom of the dish. Anyway, at 72 h the number of spheroids anchored on top of 5 μm PLGA fabric was lower than the one observed on the 20 μm PLGA fabric, eventually suggesting a stronger attachment and interaction of the stem cell spheroids with the latter configuration. Hence, spheroids seemed to adhere less stably to the 5 μm PLGA fabric. Moreover, the perimeter did not change over time. Interestingly, spheroids tend to shrink their shape and become more spherical with an aspect ratio tending to one, as compared to the more elongated shape observed at 24 h. Over time, the survival of hADSC spheroids was not compromised, as compared to that of the 2D collagen type I layer, despite them being organized in a 3D assembly, which better represents natural tissue organization, but tend to decrease the supply of nutrients and the elimination of metabolic by-products [27].

Spheroid formation is induced via cytoskeleton rearrangement. Indeed, actin filaments forming the cytoskeleton are responsible for mechanical support and for dictating the cells’ shape, which are often critical to their functions [38]. In 2D culture systems, either a 2D Petri dish or a flat collagen layer, the actin cytoskeleton formed linear filaments within the body of the cells. Conversely, the peculiar surface geometry of the microfilament network imposes structural constraints on the cell, whereby the cytoskeleton forms and reorganizes to confer a 3D cell shape with multidirectional actin arrangements in nonlinear structures. Importantly, morphological variations in stem cells acquired through cell–substrate and cell–cell interactions could influence their paracrine activity [39]. However, the influence of materials on cells’ behavior is intricate, and less is known on how cells should be handled to achieve the optimal cell function, and thereby, a therapeutic outcome in vivo [40]. Anyway, based on our preliminary study investigating the secretome of hADSC cultured on different substrates, we found that the microfilament network geometry provides a unique microenvironment that is able to enhance hADSC paracrine secretion.

Notably, the highest rate of VEGF secretion was found for the 5 μm PLGA fabric, demonstrating its potential to promote angiogenesis [41,42]. Moreover, it has emerged that highest amount of GCSF was found for the 5 μm PLGA fabric involved in stem cell mobilization. Thus, all three microstructure networks facilitated, with statistically significant differences, the secretion level of TIMP-2 relative to that in the 2D collagen layer condition. Conversely, TIMP-2 secretion was not upregulated in the collagen scaffold, making collagen a less useful substrate in these application since TIMPs—as endogenous regulators of the metalloproteinase (MMPs), which play a pivotal role in all stages of wound healing via modifying the wound matrix, thereby allowing for cell migration and tissue remodeling—may be fundamental to wound resolution [43,44,45]. The result that was just mentioned could be an another advantage of the proposed PLGA scaffolds.

Furthermore, the fabrics seemed to ameliorate—although in a not statistically significant fashion—the secretion of TFG-β1 (5 μm fabric), IL-10, RANTES (5 and 20 μm fabrics), participating in immune modulation and chemotaxis processes. A similar effect was observed for both FGF-family and EGF (10 μm fabric) that are implicated in stem cell proliferation and cell survival, respectively. Indeed there is a trend for the increased production of these factors on PLGA fabrics as compared to those of the collagen scaffold. Finally, despite the lack of other statistically significant differences with the collagen layer, the induction of proper spheroid formation using the PLGA microstructured fabrics presented—without any type of induction (i.e., active molecules)—is crucial since it has been reported that the 3D microenvironment (i.e., spheroids) offers a cellular niche that reproduces the native tissue, in which the cells maintain their natural stemness, as compared to those of the traditional two-dimensional (2D) in vitro models [46]. The side-by-side comparison with conventional substrates for cell culturing (Petri dish) and collagen layers (which can be considered a sort of gold standard for cell implantation [4]) showed that PLGA fabrics (5 and 10 µm) can significantly favor the secretion of the VEGF factor: the master regulator of vascular growth. Additionally, only a few other minor changes were observed between the PLGA fabrics and the Petri dish and collagen scaffold. Therefore, considering that plastic dish cannot be implanted and that PLGA fabrics can be further optimized and derivatized to expose specific cell adhesion molecules, these preliminary results are encouraging since all these processes are of particular interest in tissue regeneration and repair.

## 4. Materials and Methods

### 4.1. Materials

Polydimethylsiloxane (PDMS) (Sylgard 184) was purchased from Dow Corning (Midland, MI, USA). Poly-(vinyl alcohol) (PVA), poly-(lactic-co-glycolic) acid (PLGA) (50:50), Rhodamine B (RhB), and Acetonitrile (ACN) were obtained from Sigma Aldrich (St. Louis, MO, USA). Curcumin (CURC) was acquired from Alfa Aesar (Haverhill, MA, USA). Collagen Cell Carrier (CCC 10 × 10, geklebter Rand) was obtained from Viscofan Bioengineering (Weinheim, Germany). High-glucose Dulbecco’s modified Eagle’s Minimal Essential Medium (DMEM), heat-inactivated fetal bovine serum (FBS), Penicillin, Streptomycin, L-Glutamine solution, Dulbecco’s Phosphate-Buffered Saline (PBS), Trypsin-EDTA solution, Paraformaldehyde (PFA), and Tumor Necrosis Factor-α (TNF-α) were acquired from Sigma Aldrich (St. Louis, MO, USA). Human Recombinant Basic Fibroblast Growth Factor (bFGF) was obtained from Merck (Darmstadt, Germany), whereas Trypan Blue Stain (0,4%) was purchased from GIBCO (Invitrogen Corporation, Giuliano Milanese, Milan, Italy). 2′-[4-ethoxyphenyl]-5-[4-methyl-1-piperazinyl]-2,5′-bi-1H-benzimidazole trihydrochloridetrihydrate (Hoechst 33342) and 2 mg/mL Bovine Serum Albumin Standard Ampules were derived from ThermoFischer (Waltham, MA, USA). The Bicinchoninic Protein Assay kit (BCA) was obtained from Euroclone (Pero, Milan, Italy). Cytokine Array-Human Cytokine Antibody Array (Membrane, 80 targets), PE anti-CD44 antibody, FITC anti-90/Thyl antibody, and APC Anti-CD105 antibody were acquired from Abcam (Cambridge, UK). FITC Annexin V/Dead Cell Apoptosis kit, Alexa Fluor 488 Phalloidinwere sourced from Invitrogen (Waltham, MA, USA).

### 4.2. Fabrication of the PLGA Microstructured Fabrics

A soft lithography fabrication approach was employed to realize a regular network of PLGA microscopic filaments arranged as warps and wefts, presenting a rectangular cross section and perpendicularly intersecting with pitch distances of 5, 10, and 20 μm. The fabrication technique involved multiple sequential steps. First, a silicon master template was generated via direct laser writing (DLW), which engraves a matrix of square pillars on a 5-inch silicon wafer, whose edge length uniquely identifies the microfilament arrangement, pitch distance (5, 10, or 20 μm), and cross section. In the 5 and 10 μm network configurations, the width of the microscopic filaments (i.e., the distance between adjacent square pillars) is 5 μm, while for the 20 μm network configuration, this distance reduces to 3 μm. For all the configurations, the thickness of the microscopic filaments (i.e., the height of the pillars) is 5 μm. Second, a polydimethylsiloxane (PDMS) solution was deposited over the silicon template. Third, after 4 h of polymerization at 60 °C, the PDMS template, which reproduces the negative geometry of the original silicon template, was replicated into a sacrificial template via a PVA solution (3.5% w/w in water) being poured on top of it. After 1.5 h at 60 °C, the polymerized PVA template displayed the same matrix of pillars as the original silicon master template did. Fourth, a polymeric paste of PLGA was uniformly dispersed to accurately fill up the ridges between the matrix of pillars over the PVA template. Fifth, following solvent evaporation, the PLGA microfilament network was released via dissolving the PVA microlayer in water. Note that the three microfilament network configurations present different pillar and ridge sizes, and thus, require different amounts of PLGA for the same fabricated area. Specifically, the 5, 10, and 20 μm networks were obtained by using 24.5, 18.2, and 5.0 mg of PLGA, respectively. For all the configurations, PLGA was first dissolved in acetonitrile (ACN), and then spread over the whole PVA template.

To prepare RhB-loaded microscopic filaments, 2 μL of an RhB solution (2 mg/mL in ACN) was added to the PLGA paste before spreading it on top of the PVA template. To prepare curcumin-loaded microscopic filaments, 25 μL and 15 μL of a curcumin solution (10 mg/mL in ACN) were added to the PLGA paste before spreading it on the PVA template for the 5 and 10 μm configurations and for the 20 μm configuration, respectively.

### 4.3. Morphological Characterization of the Microfilament Networks

Scanning electron microscopy (SEM, JSM 6490, JEOL, Milan, Italy) was used to image all the microfilament network configurations. A portion of the PLGA network was placed on a silicon template and uniformly sputter coated with 10 nm of gold to increase the contrast and avoid damaging the sample. An acceleration voltage of 10 kV was employed for SEM imaging. Additionally, confocal microscopy (Nikon A1, Dexter, MI) was used to study the structure of all the network configurations, as well as the uniform distribution of the loaded molecules. To this end, CURC was considered for its natural green fluorescence, which under a fluorescent microscope was easily observed and quantified.

### 4.4. Biophysical Characterization of the hADSC Aggregates

Human Adipose-Derived Stem Cells (hADSC) were used to perform all the in vitro experiments. Cells were provided by the University of Navarra, Spain (Laboratory of Cell Therapy, Foundation for Applied Medical Research), and cultured at 37 °C in 5% CO_2_ in high-glucose DMEM supplemented with 10% FBS, 1% Penicillin–Streptavicin, 1% L-Glutamine, and bFGF (1 ng/mL), which from now on, is named the culture medium. For all the microfilament network configurations, 5 × 10^5^ cells were seeded on top of two polymeric networks, which were located sufficiently apart one from the other and placed on the bottom of a Petri dish. Each microfilament network had a size of 2.5 × 2.5 cm^2^, which was derived via precisely cutting the original 4 × 4 cm^2^ large network released from the PVA template with a surgical scalpel. Cells were carefully positioned in separate drops and homogeneously distributed all over the surface of the microfilament network. A few minutes after seeding, the culture medium wetted the entire surface of the substrates, and the PVA substrate started to dissolve, releasing the final PLGA microfilament network. Approximately 8 h later, the microfilament network was covered with the culture medium to prevent the dehydration of cells throughout the analysis and relocated into another Petri dish, containing a fresh culture medium, to remove the unbound cells or the ones that adhered at the bottom of the initial Petri dish.

The interaction between hADSC and the PLGA network was examined. Cell nuclei were stained with Hoechst 33342 to accurately identify the cells distributed in 3D spheroidal-like structures. The physical features of these cell aggregates were investigated using a fluorescent microscopy (Leica 6000, Wetzlar, Germany) at different time points, namely 24, 48 and 72 h, and acquiring images over the entire network. The analysis resulted in a collection of multiple fluorescent images, which were first binarized, and then thresholded via the ‘Default’ threshold method using ImageJ. These steps converted the image into binary image masks that defined groups of pixels as objects. Based on these masks, the perimeter and aspect ratio of all the objects of interest (corresponding to the cell aggregates) were quantified. The perimeter was measured as the length of the contour of each object detected in the images, whereas the aspect ratio as the ratio between the major and minor axes of the same contour.

### 4.5. Isolation of Single hADSC from Aggregates

At predetermined time points, after gently washing them with PBS, hADSC were incubated with a Trypsin-EDTA solution for 10 min. The spheroids detached from the microfilament network were further dissociated into single cell suspensions via gently pipetting the solution to avoid cell damage for 5 min. The cell suspension was collected in PBS, filtered using 40 µm cell strainers, centrifuged for 6 min at 200× *g*, re-suspended in culture medium, and counted using an automated cell counter (ChemiDoc MP, BIORAD). As a comparison, a similar protocol was applied on hADSC monolayers formed on a regular plastic dish (2D Petri dish) and an ultra-thin collagen scaffold (2D collagen layer) made of pure collagen type I fibers, which was provided by Naturin-Viscofan (Germany). The resulting suspensions were then processed for cell viability study.

### 4.6. Cell Viability of hADSC

The viability of hADSC was determined using an Annexin V-FITC Apoptosis Detection Kit and flow cytometry. After washing with PBS, individual cell suspensions (10^6^ cells/mL) were centrifuged and re-suspended in 1× Annexin Binding Buffer, which was prepared by adding 1 mL of 5× Annexin Binding Buffer to 4 mL deionized water. The single cell suspensions were incubated for 15 min at room temperature (RT) with 1.25 μL of Annexin V-FITC and Propidium Iodide (PI, at 100μg/mL). The working solution was prepared via diluting 1 mg/mL PI stock solution into 45 μL of 1× Annexin Binding Buffer. Specifically, 1 μL of 100μg/mL PI working solution was added to each 100 μL of cell suspension (1 × 10^5^ cells). After the incubation period, 400 μL of 1× Annexin Binding Buffer was added to each sample, which were immediately stored on ice and vortexed right before analysis. Flow cytometry was performed using an FACS ARIA (Becton Dickinson, Franklin Lakes, NJ, USA), and data were analyzed using FACSDiva 9.0.1 software. The experiments were performed independently at least four times.

### 4.7. Cytoskeleton Organization of hADSC

Cells were stained with Hoechst 33342 before being seeding on top of the microfilament networks to avoid any dye percolation into the PLGA structure. Additionally, to increase the imaging contrast, the red fluorescent molecule, Rhodamine B (RhB), was distributed within the PLGA network. The resulting microfilament network, which was still supported by the PVA microlayer, was cut in 1 cm^2^ pieces and placed at the bottom of the wells in an 8-well plate. Similarly, 1 cm^2^ of flat collagen scaffold was positioned in each well. Cell suspension (10^4^ cells/200µL) was seeded on top of each scaffold for 24 h. Then, cells were repeatedly washed with cold PBS (3 times per 5 min) and fixed with 0.4% paraformaldehyde (PFA) at RT for 1 h. Then, cells were washed again with PBS (3 times per 5 min) and permeabilized using 0.1% Tryton X-100 in PBS for 20 min. Finally, cells were stained with 5 μL of Phalloidin-488, which binds to cell actin, in PBS supplemented with 1% Bovine Serum Albumin (BSA) and 0.01% Tryton X-100 for 1 h.

Confocal images were used to quantify the percent of nonlinear and linear actin filaments within the cell cytoskeleton for all the configurations. After selecting the channel of interest via actin staining, all the images were firstly binarized; then, a threshold was generated using the ‘Default’ threshold method in ImageJ. These steps converted the images into binary image masks that defined groups of pixels as objects. Using ‘Analyze Particle’ function of ImageJ, the analysis parameters were set for size (pixel^2^): 0-infinity; circularity: 0–1. Based on this selection, the entire collection of objects was quantified and classified as concerns the aspect ratio, which was defined as the ratio among the major axis and minor axis of the object of interest. Finally, we arbitrarily established linear actin as the objects with an aspect ratio larger than 5 and nonlinear actin as the objects with an aspect ratio smaller than 5.

### 4.8. Secretome Analysis for hADSC

The study was performed in a model of inflamed microenvironment to observe any paracrine activity of hADSC depending on the surrounding microenvironment and the types of scaffolds used. A total of 5 × 10^5^ hADSC were cultured under controlled conditions (37 °C in 5% CO_2_), suspended in culture medium, and seeded on top of different substrates. After an 8 h incubation period, all the cell substrates were moved to a new well plate that already contained 10 mL of DMEM supplemented with 2% FBS, 1% L-Glutamine, 1% Penicillum–Streptavicin, and TNF-α (10 ng/mL), which from now on, is referred to as the pro-inflammatory culture medium. Supernatants were collected at 48 h after seeding, and cytokines, chemokines, and growth factors released by the hADSC were analyzed via cytokine/antibody array (Human Cytokine Antibody Array, Membrane, 80 Targets, ab133998) according to the manufacturer’s instructions. The arrays could assess 80 different factors simultaneously. Briefly, supernatants were first characterized in terms of total protein via a BCA assay (Bicinchoninic Protein Assay kit, Euroclone) so that we used the same amount of protein for every condition. Then, each supernatant was added to a corresponding membrane provided in the kit and kept overnight at 4 °C. The day after, following several washes with Wash Buffer I and Wash Buffer II, biotin-conjugated anti-cytokines antibody was added to all the membranes at RT for 2 h. This was followed by incubation with HRP-Conjugated Streptavidin at RT for 2 h. Finally, all the membranes were analyzed using the chemiluminescent method (ChemiDoc MP Imaging System by Bio-Rad). By using ImageJ, the luminescence associated with each spot on the membrane was quantified: higher luminescence levels are associated with higher amounts of the cytokine, chemokine, or growth factor of interest. The baseline, which is derived as the average value of the negative control spots, was subtracted from all the measured quantities. Finally, all the obtained values were normalized according to the average value of the positive control spots.

### 4.9. Statistics Analysis

All the statistical tests were performed using Graph Pad Prism 8.0. Values are presented as mean ± standard deviation (SD). Each experiment was repeated multiple times independently (≥3). Statistically significant differences among experimental groups were evaluated with an ordinary one-way ANOVA test using a Tukey’s Multiple Comparison Test as post hoc test.

## 5. Conclusions

Taking advantage of the accurate control of the size, shape, surface proprieties, and mechanical stiffness that could be obtained by using the top-down fabrication technique, three different PLGA fabric configurations, namely 5 μm, 10 μm, and 20 μm, were realized. All the three fabric configurations successfully established an intimate interaction with stem cells favoring their organization in 3D spheroidal-like structures, without affecting their viability. The investigations of the cell cytoskeleton architecture demonstrated that the fabric configuration influenced the arrangement of actin fibers favoring nonlinear organization. In addition, it was shown that the 5, 10, and 20 μm fabrics can enhance the secretion of several factors involved in key biological processes favoring injury resolution, including angiogenesis, ECM remodeling, and stem cell homing. Finally, these results highlight the possibility of modulating stem cells biological activities (i.e., angiogenesis) via using only the geometrical features of the scaffold and without using any type of active molecule. Therefore, although additional biological characterizations will be needed in future studies to assess the actual biocompatibility of the proposed PLGA microstructured fabrics, this study could pave the way for the realization of a novel, bio-functional, and tunable scaffold in regenerative medicine.

## Figures and Tables

**Figure 1 ijms-24-10123-f001:**
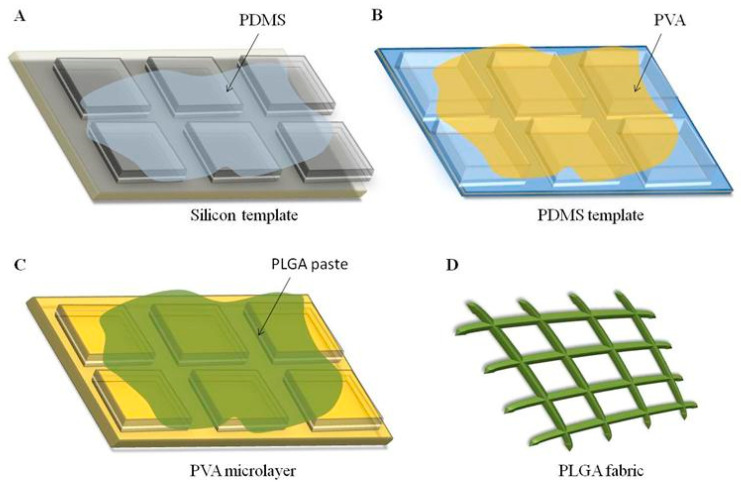
Realization of PLGA microstructured fabrics. A top-down fabrication approach relying on multiple, sequential steps was used to ensure a precise control of the network of PLGA microscopic filaments. (**A**). Silicon master template (grey), whose ridges reproduce the actual geometry of the PLGA microscopic filaments and their spatial arrangement, on which a solution of PDMS was cast (light blue). (**B**). Upon curing, the PDMS template was coated with an aqueous solution of PVA (yellow), leading to a firm PVA template after drying. (**C**). The ridges of the PVA template were carefully filled with a polymeric paste (green), obtained via dissolving PLGA into a highly volatile organic solvent. (**D**). Upon exposure to water, the sacrificial PVA template dissolves and releases in solution the network of PLGA microscopic filaments (PLGA microstructured fabric).

**Figure 2 ijms-24-10123-f002:**
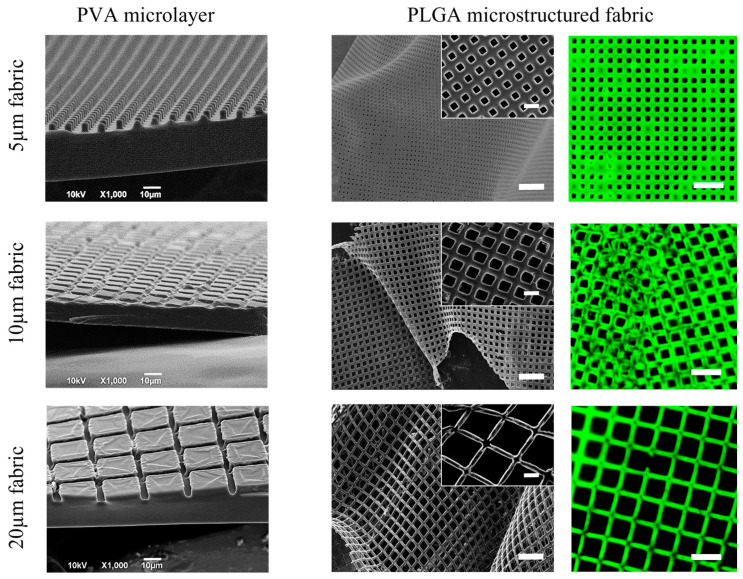
Geometrical characterization of the three PLGA microstructured fabrics. Scanning electron microscopy images of the sacrificial PVA templates (scale bar: 10 μm; **left column**). Scanning electron microscopy documenting the mechanical flexibility of the PLGA fabrics (scale bar: 50 μm; inset scale bar: 10 µm; **central column**). Fluorescent microscopy images demonstrating the uniform distribution of the natural green, fluorescent molecule, curcumin, within the PLGA microscopic filaments of the fabric (scale bar: 30 μm; **right column**). 5, 10, and 20 μm PLGA fabric configurations are presented in the top, central, and bottom rows, respectively.

**Figure 3 ijms-24-10123-f003:**
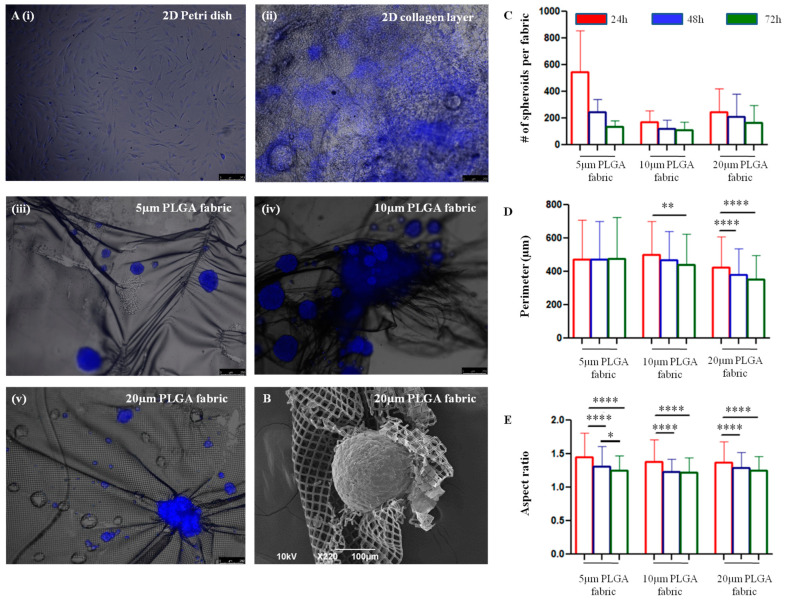
Analysis of stem cell organization on PLGA fabrics. (**A**). Fluorescent microscopy images showing the distribution and adherence of hADSC on a 2D Petri dish (**i**) and 2D Collagen type I scaffolds (**ii**), in contrast to the formation of spheroidal aggregates on 5 μm (**iii**), 10 μm (**iv**), and 20 μm PLGA fabrics (**v**). Cell nuclei were stained in blue with Hoechst 33342. (**B**). Magnified scanning electron microscopy image of an hADSC spheroid surrounded and wrapped around 20 μm PLGA fabric. (**C**–**E**). Number, perimeter, and aspect ratio of hADSC spheroids formed on different PLGA fabric configurations at predetermined time points, namely 24, 48, and 72 h. Data are presented as mean ± SD (*n* ≥ 3). Differences are considered to be statistically significant when *p* < 0.05 (*), *p* < 0.01 (**), and *p* < 0.0001 (****).

**Figure 4 ijms-24-10123-f004:**
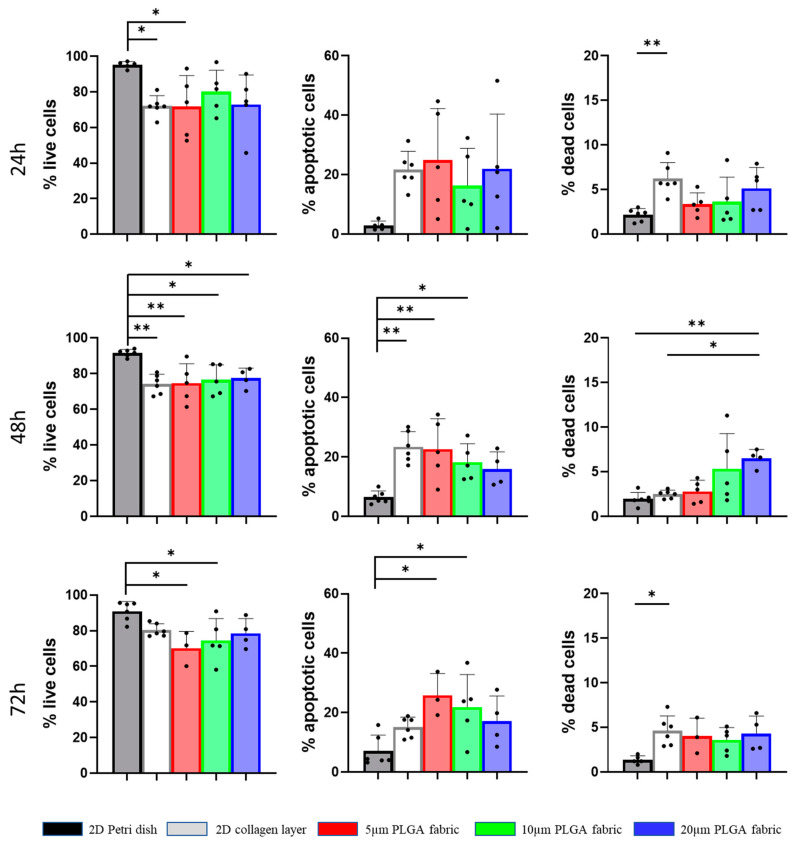
Viability of stem cells on PLGA microstructured fabrics. Flow cytometry analyses showing the percentage of live, apoptotic, and dead hADSC cultured at predetermined time points on Petri dishes (dark grey bar); 2D collagen type I scaffolds (light grey bar); 5 μm PLGA fabric (red bar); 10 μm PLGA fabric (green bar); 20 μm PLGA fabric (blue bar). Data are presented as mean ± SD (*n* ≥ 3). Differences were considered to be statistically significant for *p* < 0.05 (*) and *p* < 0.01 (**).

**Figure 5 ijms-24-10123-f005:**
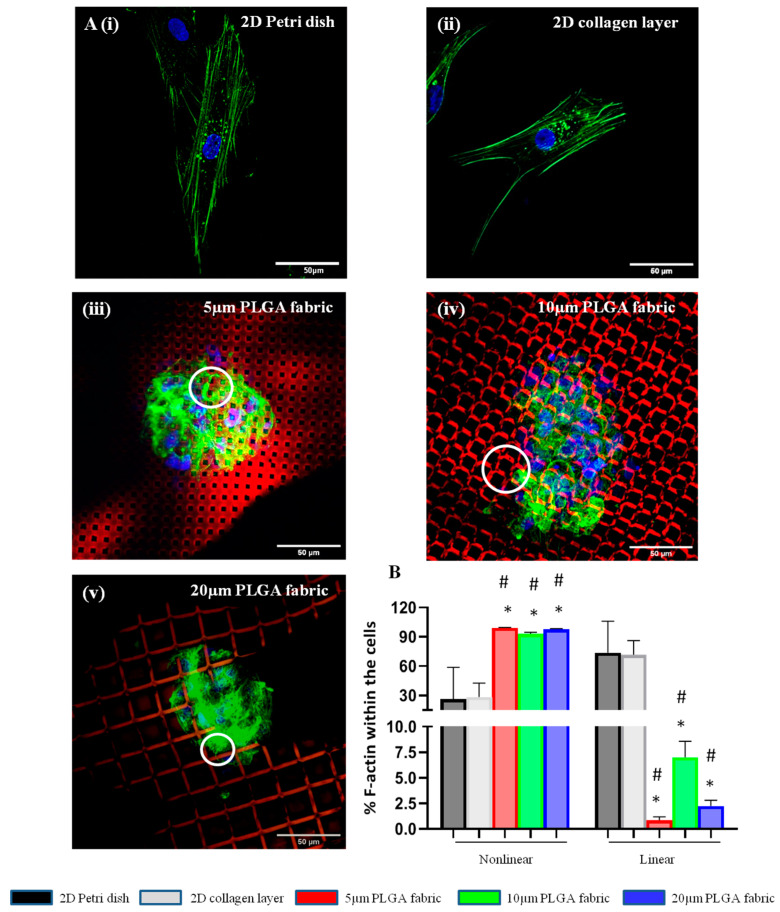
Actin filament organization of hADSC cultured on PLGA microstructured fabrics. (**A**). Confocal microscopy images showing the linear organization of the actin filaments in hADSC spreading and adhering on 2D Petri dishes (**i**) and 2D collagen type I scaffolds (**ii**), as opposed to the more complex organization observed for 5 (**iii**), 10 (**iv**), and 20 µm PLGA fabrics (**v**). White circles indicate the ring-like structures at the points of anchorage of hADSC with the three PLGA microstructured filaments (**B**). Quantification of the actin organization structure within the hADSC (* represents a statistically significant difference between this and the 2D Petri dish: *p* < 0.05; # represents a statistically significant difference between this and the collagen type I layer: *p* < 0.05) (*n* > 3).

**Figure 6 ijms-24-10123-f006:**
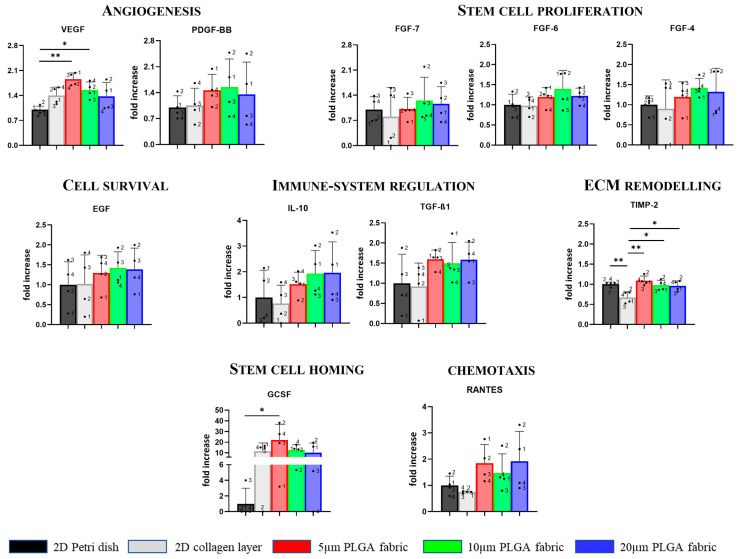
Secretome analysis for TNF-α stimulated stem cells on PLGA microstructured fabrics. Cytokines’ expression levels among the different scaffolds at 48 h. Data are presented as mean ± SD (*n* ≥ 3). Differences are considered to be statistically significant for *p* < 0.05 (*) and *p* < 0.01 (**).

## Data Availability

The data presented in this study are available on request from the corresponding author.

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
