# Peer review of "Microstructured Polymeric Fabrics Modulating the Paracrine Activity of Adipose-Derived Stem Cells"

_ijms, 2023, doi:10.3390/ijms241210123_

Round 1
Reviewer 1 Report
The work described in this manuscript is centered around the development of implantable PLGA mesh fabric-based scaffolds for maintenance of secretory functions of adipose derived stem cells. The authors demonstrate that spheroid formation is superior in these flexible scaffolds in comparison to collagen scaffolds or a petri dish and aver that the paracrine activity is therefore enhanced.
1. The premise behind the study that flexible films will support spheroid formation is not novel and a whole range of materials both natural and synthetic (gelatin, chitosan, hyaluronic acid, graphene, etc.) have previously been explored. The authors may include a deeper discussion with additional references as to the advantages of their scaffolds over existing materials as well as other biomaterial formats such as the use of particles to support spheroid formation. If the advantage of the PLGA scaffolds is, as the authors state, improved mechanical properties and slower degradation, these aspects must be characterized.
2. While a statistical comparison of spheroid formation and size was performed for each group as a function of time, it is not clear whether there are differences between the groups at each of these time points or overall.
3. The authors suggest that the drop in viability in the spheroids is due to the lack of nutrients to the core. This may be verified by correlating viability with the size of the spheroid. Moreover, it may be worth conducting viability, proliferation and metabolic activity assays by staining to assess spatial gradients in cell behavior as a function of nutrient access.
4. The authors may provide further discussion over the fact that only two secretory factors were significantly upregulated in the flexible scaffold model over plastic and there was no significant difference in comparison with collagen scaffolds. TIMP-2 is atleast comparably upregulated in the petri dish as in the PLGA scaffolds which suggests that it may be a result of the absence of native cell binding sites in synthetic scaffolds.
Author Response
please see the attachemnt

Reviewer 2 Report
This study developed a regular network of microscopic PLGA filaments. On the PLGA fabric, hADMSC spheroidal-like structure could be formed. This PLGA network could preserve cell viability, and favor a nonlinear actin organization. Thus, PLGA fabric would be a promising biodegradable scaffold for hADMSC tissue integration. However, Nevertheless, there are some comments necessary to be concerned before the article could be accepted for publication.
(1) many other research has reported microstructured polymer fabrics and its application in tissue engineering, such as PET, PCL, PLA, PGA, and PLGA fabrics. In the Introduction, the author should add more comments in this field.
(2) Could Figure 3A be revised with sub-capture title (such as 3A (i), (ii), (iii), (iv), and (v))?
(3) Figure 3B showed the SEM image of hADMSC spheroid surrounded and wrapped by 20 um PLGA fabric. Did this image was taken after culture 24h? What are the hADMC spheroid when the other PLGA fabric (5 um, 10 um)?
(4) As shown in Figure 3C-E, the # of spheroids per fabric, perimeter and aspect ratio decreased as the culture time increased (24h, 48h, 72h). Could the author give more discussion about the mechanism? What will it happen if the culture time increase 1 week?
(5) In the Figure 4, the cell viability on PLGA microstructured fabrics was lower than that on 2D Petri dish. The author gave explanation that the different spatial cell organization is expected to affect the distribution of nutrients to the core of the hADMSC aggregates. However, the cell viability (% live cells) on PLGA microstructured fabrics was decreased when the culture time was increased from 24h to 72h. Why?
(6) There was no label (A, B) in Figure 5, which caused confuse about it. The author should revise it.
Round 2
Reviewer 1 Report
The authors have sufficiently addressed all concerns and the manuscript may be accepted in this present form
Reviewer 2 Report
The authors have addressed my concerns. It could be accepted.